# Home Cooking Is Related to Potential Reduction in Cardiovascular Disease Risk among Adolescents: Results from the A-CHILD Study

**DOI:** 10.3390/nu12123845

**Published:** 2020-12-16

**Authors:** Yukako Tani, Takeo Fujiwara, Aya Isumi, Satomi Doi

**Affiliations:** 1Department of Global Health Promotion, Tokyo Medical and Dental University (TMDU), Yushima, Bunkyo-ku, Tokyo 113-8519, Japan; tani.hlth@tmd.ac.jp (Y.T.); isumi.hlth@tmd.ac.jp (A.I.); doi.hlth@tmd.ac.jp (S.D.); 2Japan Society for the Promotion of Science, Kojimachi, Tokyo 102-0083, Japan

**Keywords:** adolescents, home cooking, meal preparation, blood pressure, serum cholesterol, overweight

## Abstract

This study aimed to investigate the association between the frequency of home cooking and cardiovascular disease risk among Japanese adolescents. We used cross-sectional data on adolescents from the 2018 Adachi Child Health Impact of Living Difficulty study, which targeted junior high school students aged 13–14 years in Adachi, Tokyo, Japan. Frequency of home cooking by 553 caregivers was assessed via questionnaire and classified as high (almost daily), medium (4–5 days/week), or low (≤3 days/week). Cardiovascular disease risk factors included blood pressure, serum cholesterol (total, LDL, and HDL), hemoglobin A1c, and body mass index. Multiple linear regression analysis revealed that adolescents exposed to a low frequency of home cooking showed higher diastolic blood pressure (β = 3.59, 95% confidence interval [CI]: 0.42 to 6.75) and lower HDL cholesterol (β = −6.15, 95% CI: −11.2 to −1.07) than those exposed to a high frequency of home cooking, adjusting for adolescents’ sex, household income, and parental comorbidity. Future studies are needed to clarify the causal relationship and mechanisms through which home cooking influences adolescents’ cardiovascular health.

## 1. Introduction

Since the late 20th century, sociodemographic changes in developed countries such as an increasing percentage of families in which both parents work have decreased the time available for home cooking, and people have come to depend on meals away from home [1,2,3]. In 2000, compared with the 1960s, the percentage of daily energy consumed from home food sources had dropped by about 25% in the United States [1]. Recently, more than 50% of American adults reported consuming meals away from home at least three times per week [4]. In Japan, eating out is widespread among younger people, and household expenditure on pre-cooked food increased by 26% from 1993 to 2015 [5]. Because meals away from home, such as fast food, restaurant meals, and convenience food, are higher in energy density, fat, and sodium and lower in protective nutrients than home-cooked foods, these meals are associated with poor diet quality and increased energy intake in both adults and children [6,7,8,9,10,11,12,13]. These nutritional differences are reflected in the association of meals away from home with metabolic risk outcomes among adults [4,14,15].

There are increasing calls for a return to home cooking to prevent poor dietary habits, which induce many diseases in adulthood, including cardiovascular diseases [16]. A systematic review has reported the dietary benefits of home cooking, including greater consumption of healthier food groups such as fruits and vegetables, enhanced nutrient intake, and improved adherence to healthy dietary recommendations such as the Healthy Eating Index and the Mediterranean diet [17]. Although studies examining outcomes beyond diet are limited, a recent population-based study among adults in the United Kingdom showed that regularly eating home-cooked meals was associated with better outcomes in terms of several markers for cardiovascular diseases, including adiposity, cholesterol, and hemoglobin A1c (HbA1c) [18].

Cardiovascular disease risk is defined by the presence of any three of the following five risk factors: hyperglycemia, overweight or obesity, elevated triglycerides, low high-density lipoprotein (HDL) cholesterol, and elevated blood pressure (BP) [19]. Cardiovascular diseases are the leading cause of death globally [20], and a significant slowdown in the decline of cardiovascular disease mortality has been reported in high-income countries including Japan in recent years [21]. Cardiovascular risk starts in childhood, especially adolescence [19]. Thus, to prevent future cardiovascular diseases, it is important to examine how cardiovascular risk in adolescence is associated with home meal preparation, which may be a modifiable preventive factor. One cross-sectional study reported that higher HDL cholesterol levels were observed among adolescents in the United States when family dinners were purchased from take-out sources at least weekly [22]. However, this previous study did not account for household income, which is considered an important confounding factor. The study also used a convenience sample drawn from existing cohort studies and from local communities, which raises concerns regarding sampling bias. Additionally, to consider the effects of both eating out and consuming prepared meals at home, it is necessary to examine the frequency of cooking at home. Furthermore, to our knowledge, no previous studies have examined this association among Asian adolescents. Risk factors for cardiovascular disease vary by region, and compared with individuals in Western countries, Asians tend to have a lower body mass index (BMI) but a higher prevalence of diabetes [23]. Therefore, it is important to provide evidence on risk factors for cardiovascular disease in Asians, specifically.

One of the most important dietary modifications related to cardiovascular disease is vegetable intake [24]. Inadequate vegetable intake is highly prevalent among all age groups, especially adolescents, who eat half of the recommended amount of vegetables [25]. Habits regarding breakfast, the first meal of the day, have a critical role in dietary nutrient adequacy [26]. A recent systematic review showed that skipping breakfast was associated with metabolic syndrome, including worse lipid profiles, blood pressure levels, and insulin resistance [27]. Nevertheless, the prevalence of breakfast skipping among adolescents is high and rising [26,27]. Additionally, cardiovascular disease risks, such as hypertension, hyperglycemia, and dyslipidemia, are becoming increasingly common in adolescents with obesity [28]. A previous study in Japan showed that children exposed to a low frequency of home cooking had lower frequencies of vegetable consumption and breakfast intake and tended to be obese [29]. Therefore, adolescents who eat less home-cooked food may be vulnerable to developing unhealthy dietary behaviors, which, in turn, may be linked to cardiovascular disease risk.

The purpose of this study was to examine the associations between the frequency of home cooking and indicators of cardiovascular disease risk among adolescents in Japan. Moreover, we estimated the proportion of the association between home cooking and cardiometabolic indicators, which was mediated by potential mediators including adolescents’ dietary behaviors. We used vegetable intake and breakfast intake as proxies for diet quality [30,31,32].

## 2. Materials and Methods

### 2.1. Study Design and Subjects

We used cross-sectional data collected in 2018 in the Adachi Child Health Impact of Living Difficulty (A-CHILD) study, which was established in 2015 to evaluate the determinants of health among children in Adachi, Tokyo, Japan [33]. This study covered seven public junior high schools established by Adachi City. In 2018, self-report questionnaires were distributed to 676 13- to 14-year-old adolescents in their second year of junior high school. Adolescents and their caregivers completed these questionnaires at home and then returned the completed questionnaires to the school. A total of 632 pairs of adolescents and their caregivers returned the questionnaires (response rate: 94%), and 583 provided informed consent and returned all questionnaires (consented rate: 92%). Over 95% of the complete responses from consenting participants could be linked with adolescents’ school medical examination data, including BP, serum cholesterol (total, low-density lipoprotein (LDL), and HDL), HbA1c, height, and weight (*n* = 554 for BP and blood sampling tests and 569 for height and weight). These medical examinations, “child lifestyle-related health checkups”, were conducted by Adachi City for all students in the second grade of junior high school [34]. The medical examinations were carried out at school, and blood sampling was optional. The present analyses were conducted using data on 553 pairs of adolescents and their caregivers for BP and serum cholesterol and 568 adolescent–caregiver pairs for BMI, after excluding a participant who did not complete the questions related to home-cooking status. Because HbA1c is affected by anemia, the analysis of HbA1c was performed after excluding adolescents who were anemic, which was defined as having a hemoglobin level < 12 g/dL (*n* = 22) [35]. The A-CHILD protocol and use of the data for this study were approved by the Ethics Committee at the Tokyo Medical and Dental University (No. M2016-284).

### 2.2. Outcomes

Adolescents underwent school medical examinations, including anthropometric measurements, blood sampling tests, and BP measurements. BP was measured twice using a sphygmomanometer on the right arm while sitting, and the averages of the two measurements of systolic BP (SBP) and diastolic BP (DBP) were used for the analysis. Blood samples were obtained from the adolescents by venipuncture. Plasma samples were measured for total cholesterol, LDL cholesterol, HDL cholesterol, and HbA1c. Height and weight were measured according to a standardized protocol [36]. Specifically, height was measured to the nearest 0.1 cm using a portable stadiometer, and weight was measured to the nearest 0.1 kg on a digital scale, without shoes and wearing light clothing. BMI was calculated by dividing the child’s weight in kg by the square of their body height in meters and is expressed as a *z*-score representing the deviation in SD units from the mean of the standard normal distribution of BMI specific to age and sex, according to the World Health Organization Child Growth Standards. Overweight was defined as having a BMI greater than 1 SD above this mean [37]. For the analysis, continuous values were used for BP, serum cholesterol (total, LDL, and HDL), HbA1c, and BMI, and binary values were used for overweight status.

### 2.3. Home-Cooking Frequency

Home-cooking frequency over the past month was assessed by asking caregivers the following question: “How many times did you or someone else in your family cook meals at home? A cooked meal is defined as a simple meal, such as a fried egg” [29]. The five response items were 6 or more days/week, 4–5 days/week, 2–3 days/week, a few days/month, and rarely. Following previous studies [29,38,39], we collapsed the responses into three frequency of home cooking categories: (i) high (6 or more days/week); (ii) medium (4–5 days/week); and (iii) low (≤3 days/week).

### 2.4. Covariates

Household annual income (<3.00, 3.00–5.99, 6.00–9.99, or ≥10.0 million JPY), caregiver’s comorbidity (mother’s and father’s diabetes and hypertension), and adolescent’s frequency of vegetable intake (twice/day, once/day, or fewer than three times/week) were assessed via caregiver reports [29]. Adolescent’s physical activity was assessed based on the frequency of physical activity for 30 minutes or more during the week (never/rarely, one to two times/week, three to four times/week, or five or more times/week) via self-report. Adolescent’s breakfast intake (every day, often, or rarely/never) was also assessed via self-report.

### 2.5. Statistical Analysis

First, Pearson’s chi-square test was used to determine differences in demographic data by home-cooking status. Second, multiple linear regression analysis was used to assess mean differences in indicators of cardiometabolic factors (BP, serum cholesterol (total, LDL, and HDL), HbA1c, and BMI) by home-cooking status. We calculated adjusted odds ratios with 95% confidence intervals (CIs) of overweight using logistic regression. The models were adjusted for adolescent’s sex, household income, and mother’s diabetes as potential confounders. Third, we conducted a mediation analysis to determine the proportion of the association between home cooking and cardiometabolic indicators that was mediated by the potential mediators. Using the Paramed package in Stata [40], we estimated the natural direct effects, controlled direct effects, and natural indirect effects of mediators after controlling for all covariates (adolescent’s sex, household income, and mother’s diabetes). The exposure was treated as a binary variable, with 0 representing the middle/high frequency of home cooking and 1 representing the low frequency of home cooking. The mediators and outcomes were treated as continuous variables. Each mediator was assessed separately to estimate its unique direct and indirect effects. All analyses were conducted using Stata, version 15 with the significance level set at 0.05.

## 3. Results

Among all adolescents, 293 (53%) were girls, 374 (68%) reported physical activity for 30 minutes or more at least once a week, and 60 (11%) were overweight (Table 1). Among the participating families, 469 (85%) cooked almost every day (high), 63 (11%) cooked 4–5 days per week (medium), and 21 (3.8%) cooked fewer than 3 days per week (low). In terms of household status, 58 (11%) were low-income households with annual incomes less than three million JPY, parental diabetes was present for three (0.5%) mothers and 14 (2.5%) fathers, and parental high BP was present for 34 (6.1%) mothers and 43 (7.8%) fathers. Among all adolescents, 70 (13%) consumed vegetables fewer than three times per week, and 94 (17%) did not eat breakfast every day. Among low-income households, the frequency of home cooking was low. Adolescents exposed to a low frequency of home cooking tended to consume fewer vegetables, skip breakfast, and be overweight.

The associations between frequency of home cooking and indicators of cardiovascular disease risk are shown in Table 2. Multiple linear regression analysis revealed that adolescents exposed to a low frequency of home cooking had higher DBP (β = 3.59, 95% CI: 0.42 to 6.75) and lower HDL cholesterol (β = −6.15, 95% CI: −11.2 to −1.07) compared with those exposed to a high frequency of home cooking, after adjusting for confounders (adolescent’s sex, household income, and mother’s comorbidity). Adolescents exposed to a medium frequency of home cooking were 2.67 times (95% CI: 1.37 to 5.23) more likely to be overweight than those exposed to a high frequency of home cooking. No significant association was found between low frequency of home cooking and overweight. Adolescents exposed to a low frequency of home cooking did not have a higher risk of overweight than those exposed to a high frequency of home cooking. No significant association was found between frequency of home cooking and SBP, total cholesterol, LDL cholesterol, or HbA1c.

Appendix A displays the mediation results for the hypothesized mediators, comparing adolescents exposed to a low frequency of home cooking with those exposed to a middle/high frequency of home cooking. We found evidence of mediation between home cooking and DBP through breakfast skipping (12.3%, *p* = 0.19), but we did not find evidence of mediation through BMI (2.2%, *p* = 0.58) and/or vegetable intake (−3.4%, *p* = 0.55). For the association with HDL, we observed evidence of mediation through low frequency of vegetable intake (9.7%, *p* = 0.09) and breakfast skipping (14.9%, *p* = 0.10) but not through BMI (6.3%, *p* = 0.56).

## 4. Discussion

Our study showed associations between frequency of home cooking and the cardiovascular disease risk factors of BP, serum cholesterol (total, LDL, and HDL), HbA1c, and BMI among adolescents aged 13–14 years. We found that a low frequency of home cooking was associated with higher DBP and lower HDL cholesterol, after adjusting for potential confounders. The association between frequency of home cooking and overweight was significant when comparing adolescents exposed to a medium frequency of home cooking with those exposed to a high frequency of home cooking. Adolescents’ unhealthy eating behaviors (low frequency of vegetable intake and breakfast skipping) mediated about 10% of the associations of a low frequency of home cooking with higher DBP and lower HDL cholesterol, although the adolescents’ BMI did not mediate these associations.

To our knowledge, no previous studies have examined the association between home cooking and BP in adolescents. In terms of studies on adults, a previous study of middle-aged people in the United States reported that there was no significant association between home cooking and hypertension [18]. However, this previous study defined hypertension as having a DBP of 90 mmHg or higher or an SBP of 140 mmHg or higher and had the limitation of not obtaining information on medication. Therefore, it is difficult to discuss consistency between our study and this previous study. The association between less home cooking and adolescents’ high DBP may be explained by BMI-dependent and BMI-independent pathways.

We did not find evidence of mediation through BMI, which suggests that BMI-independent pathways may operate in our setting. We found that adolescents’ breakfast skipping partially mediated the association between a low frequency of home cooking and higher DBP. Our findings showed that 85% of adolescents exposed to a high frequency of home cooking ate breakfast every day, whereas only 43% of adolescents exposed to a low frequency of home cooking ate breakfast every day (Table 1). A systematic review of the associations between breakfast skipping and cardiometabolic risk factors in adolescents showed that breakfast skipping was associated with higher BP [27]. The mechanism of this association is not well understood; however, a study using a national nutritional survey in Japan showed that overall diet quality was higher among breakfast consumers than among breakfast skippers across all age groups, including adolescents [41]. BMI-independent pathways may operate through pathways other than vegetable intake and breakfast skipping because we found that the mediation role of these eating behaviors was small. Salt intake may be another BMI-independent pathway. Sodium intake has been shown to be positively associated with BP in adolescents [42]. In Japan, most dietary sodium comes from commercially processed foods [43]. Therefore, adolescents exposed to a low frequency of home cooking may tend to consume more salt compared with those exposed to a high frequency of home cooking.

We also found that a low frequency of home cooking was associated with lower HDL cholesterol. A previous study of middle-aged adults reported that consuming home-cooked meals more than five times per week was associated with a decreased risk of a high ratio of total cholesterol to HDL cholesterol, compared with eating home-cooked meals fewer than three times per week, in a crude model [18]. In a study of adults conducted in the United States, a high frequency of fast-food meals was associated with lower HDL cholesterol (HDL was 51 mg/dL for those who ate fast food four times per week and 54 mg/dL for those who ate no fast food) [4]. Among adolescents in the United States, lower HDL was also observed when family dinners were purchased from take-out sources at least once in the past week, compared with never in the past week [22]. Our findings on HDL cholesterol were consistent with these previous studies.

In the mediation analysis, 9.7% of the indirect effect of home cooking on HDL cholesterol was shown to operate through vegetable intake, and 14.9% of this effect was shown to operate through breakfast intake. Thus, vegetable intake and breakfast consumption may partially explain the mechanism through which home cooking boosts levels of HDL cholesterol. A review of the association between diet and cardiometabolic health reported that some nutrients and foods were potentially linked to HDL cholesterol level among adolescents [44], although these relationships are still poorly understood [45]. A systematic review of randomized, controlled trials showed that the most effective foods in terms of increasing HDL cholesterol were fish, nuts, and eggs [46]. A systematic review and a recent study of middle-aged adults identified the dietary benefits of home cooking, including greater consumption of fruits and vegetables and improved adherence to healthy dietary guidelines such as the Mediterranean diet, which is rich in fish and nuts [17,18]. Therefore, the vegetable intake of the adolescents in our study may be an indicator of this population’s consumption of fish or nuts, rather than only vegetables. In terms of breakfast consumption, a systematic review showed evidence that breakfast skipping is associated with lower HDL cholesterol [27]. In Japan, compared with adolescents who skip breakfast, those who consume breakfast eat more eggs [41], a food that may increase HDL cholesterol [46]. Future studies are needed to clarify the mechanisms through which home cooking influences adolescents.

We found a U-shaped association between home cooking and overweight. One potential reason for this is that adolescents exposed to a low frequency of home cooking may not be able to consume enough energy because of skipping meals, for example, which leads to these adolescents being underweight. We found that 11% of adolescents exposed to a medium frequency of home cooking were underweight, whereas 19% of adolescents exposed to a low frequency of home cooking were underweight (Table 1). Additionally, the proportion skipping breakfast was twice as high among adolescents exposed to a low frequency of home cooking and among adolescents exposed to a medium frequency of home cooking (Table 1). Therefore, if home cooking is infrequent (in the present study, fewer than three times a week), adolescents may be at risk of becoming either overweight because of an inadequate diet or underweight because of malnutrition. However, with a small number of households reporting a low cooking frequency, we had limited power to detect the association between home cooking and overweight.

Our study has some limitations. First, because home cooking is defined as simple cooking that can include fried eggs, for instance, there is a possibility of misclassification: Even simple foods or low-quality meals cooked at home could be assigned to the high-frequency group. This may lead to an underestimation of the effect of home cooking on adolescents’ health outcomes. Second, the present study was implemented in one city in Tokyo. Given that home cooking practices can vary depending on whether the family is nuclear or multigenerational, the mother’s employment status, and the household’s urban or rural status, our results cannot be generalized. Third, we assessed cardiovascular disease risks other than BMI using only continuous values. However, for adolescents, there is no need to use cut points based on the definition of metabolic syndrome [19], and continuous values have been reported to be more reliable in predicting risk in adulthood from assessments in adolescence [47]. Furthermore, the number of adolescents whose HbA1C values corresponded to prediabetes (HbA1c ≥ 5.7) was small (*n* = 21); we therefore decided to use a continuous scale for this variable. Fourth, we lacked information on diet quality. Although we used vegetable intake and breakfast intake as proxies for diet quality, a study of detailed diet quality is needed to further elucidate the mechanisms at work here. Fifth, although we accounted for household income and parental comorbidities as confounders, other potential confounding factors such as parental education level, smoking status, and food knowledge were not adjusted. Moreover, although we assessed adolescents’ frequency of physical activity via questionnaire, we did not assess detailed physical activity levels such as activity type or intensity, which may have biased the association between home cooking and indicators of cardiometabolic factors. For example, caregivers may want to feed their adolescent children additional food to compensate for the calories expended through vigorous physical activity, and these parents may therefore cook more often. Furthermore, physical activity can have a direct impact on cardiometabolic risk factors [48,49]. Finally, the cross-sectional nature of this study means that causality could not be assessed. Longitudinal studies or randomized, controlled trials of home cooking among families with adolescents should be conducted in the future to determine the causal impact of these practices on cardiovascular risk factors.

Despite these limitations, our study has shown a protective association between home cooking and adolescent health. Although the causality is unknown, our findings suggest that caregivers may be able to promote the health of adolescents in the household by cooking at home daily. In the future, it is necessary to clarify the mechanism of this association through detailed dietary surveys and intervention studies.

## Figures and Tables

**Table 1 nutrients-12-03845-t001:** Characteristics of adolescents and caregivers enrolled in the study (*n* = 553).

	Total		Frequency of Home Cooking			*p*-Value
			High	Medium	Low	
			(*n* = 469, 85%)	(*n* = 63, 11%)	(*n* = 21, 3.8%)	
	*n*	%	%	%	%	
Adolescent’s status						
Sex						
Boy	260	47.0	46.7	49.2	47.6	0.93
Girl	293	53.0	53.3	50.8	52.4	
Physical activity						
Never/rarely	178	32.2	32.2	33.3	28.6	0.80
1–2 times/week	154	27.8	26.9	30.2	42.9	
3–4 times/week	89	16.1	16.0	15.9	19.0	
≥5 times/week	131	23.7	24.7	20.6	9.5	
Missing	1	0.2	0.2	0.0	0.0	
Body weight status (BMI for age *z* score)						
Underweight (<−1 SD)	102	18.4	19.4	11.1	19.0	0.001
Normal weight (−1 SD to +1 SD)	382	69.1	70.4	63.5	57.1	
Overweight (>+1 SD)	60	10.8	9.2	22.2	14.3	
Missing	9	1.6	1.1	3.2	9.5	
Eating behaviors						
Frequency of vegetable intake						
Twice/day	231	41.8	45.4	20.6	23.8	<0.001
Once/day	249	45.0	44.8	49.2	38.1	
Fewer than three times/week	70	12.7	9.2	30.2	38.1	
Missing	3	0.5	0.6	0.0	0.0	
Frequency of breakfast intake						
Every day	453	81.9	85.3	69.8	42.9	<0.001
Often	60	10.8	9.6	14.3	28.6	
Rarely/never	34	6.1	3.8	15.9	28.6	
Missing	6	1.1	1.3	0.0	0.0	
Household and caregiver’s status						
Household income (million JPY)						
<3.00	58	10.5	9.2	17.5	19.0	0.03
3.00–5.99	167	30.2	29.9	34.9	23.8	
6.00–9.99	181	32.7	34.5	27.0	9.5	
≥10.0	49	8.9	9.4	4.8	9.5	
Missing	98	17.7	17.1	15.9	38.1	
Mother’s comorbidity						
Diabetes	3	0.5	0.2	3.2	0.0	0.01
Hypertension	34	6.1	5.5	11.1	4.8	0.22
Father’s comorbidity						
Diabetes	14	2.5	2.3	3.2	4.8	0.74
Hypertension	43	7.8	8.1	6.3	4.8	0.77

BMI: body mass index; SD: standard deviation.

**Table 2 nutrients-12-03845-t002:** Results of linear and logistic regression analyses of indicators of cardiometabolic factors for Japanese adolescents by frequency of home cooking.

	*n*	Mean (SD)	β (95% CI)
Blood pressure			
SBP (mmHg)			
Frequency of home cooking			
High	469	108.8 (9.5)	reference
Medium	63	110.2 (9.1)	1.36 (−1.11 to 3.82)
Low	21	111.6 (8.3)	2.02 (−2.07 to 6.11)
DBP (mmHg)			
Frequency of home cooking			
High	469	62.1 (7.3)	reference
Medium	63	62.3 (6.9)	0.24 (−1.67 to 2.15)
Low	21	65.6 (7.5)	**3.59 (0.42 to 6.75)**
Cholesterol level			
Total cholesterol (mg/dL)			
Frequency of home cooking			
High	469	167.2 (26.7)	reference
Medium	63	168.6 (21.8)	1.81 (−4.94 to 8.56)
Low	21	162.1 (26.5)	−4.84 (−16.02 to 6.34)
LDL (mg/dL)			
Frequency of home cooking			
High	469	90.3 (23.1)	reference
Medium	63	93.2 (20.5)	3.64 (−2.42 to 9.69)
Low	21	93.3 (28.4)	3.15 (−6.89 to 13.18)
HDL (mg/dL)			
Frequency of home cooking			
High	469	62.2 (11.9)	reference
Medium	63	61.3 (10.7)	−1.06 (−4.13 to 2.00)
Low	21	55.9 (10.0)	**−6.15 (−11.2 to −1.07)**
HbA1c (%)			
Frequency of home cooking			
High	450	5.34 (0.20)	reference
Medium	61	5.34 (0.19)	−0.001 (−0.053 to 0.052)
Low	20	5.33 (0.14)	−0.015 (−0.103 to 0.073)
BMI (*z* score)			
Frequency of home cooking			
High	483	−0.19 (1.01)	reference
Medium	63	0.15 (1.16)	**0.30 (0.03 to 0.58)**
Low	22	−0.02 (1.05)	0.11 (−0.3 to 0.55)
Overweight: *n* (%)			OR (95% CI)
Overweight (>+1 SD)			
Frequency of home cooking			
High	483	47 (9.7)	reference
Medium	63	15 (23.8)	**2.67 (1.37 to 5.23)**
Low	22	3 (13.6)	1.28 (0.36 to 4.58)

BMI: body mass index; CI: confidence interval; DBP: diastolic blood pressure; HbA1c; hemoglobin A1c; HDL: high-density lipoprotein; LDL: low-density lipoprotein; OR: odds ratio; SBP: systolic blood pressure; SD: standard deviation. The models were adjusted for adolescent’s sex, household income, and mother’s diabetes. Boldface indicates statistical significance (*p* < 0.05).

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
