# Peer review of "Home Cooking Is Related to Potential Reduction in Cardiovascular Disease Risk among Adolescents: Results from the A-CHILD Study"

_nutrients, 2020, doi:10.3390/nu12123845_

Round 1

Reviewer 1 Report

The authors have successfully addressed my concerns. It is time to prime.

Author Response

We thank the Reviewer for their helpful comments.

Reviewer 2 Report

I reviewed the revised manuscript. Revised Manuscript is well written. All comments have been addressed and thus accepted for publication.

Author Response

We thank the Reviewer for their helpful comments.

This manuscript is a resubmission of an earlier submission. The following is a list of the peer review reports and author responses from that submission.

Round 1

Reviewer 1 Report

  1. Amount of salt intake should also be taken into consideration.
  2. Table 2 should be re-designed/structured. Frequency of home cooking in the left upper column is not correct. For example, blood pressure and Cholesterol levels are not part of Frequency of home cooking, but they are under Frequency of home cooking column.
  3. Abbreviations should be provided as footnotes for all tables.
  4. Any data on education levels of parents in household
  5. Any data on smoking status in parents 
  6. Major limitation is cross-sectional design limiting assessment of causal relationship
  7. About minor points, there are grammars and typos errors in the text. Please thoroughly check the article

Reviewer 2 Report

This work present by Tani et al investigates the association of the frequency of home cooking with the risk of cardiovascular diseases. The work was conducted properly and the statistical analysis lead to the solid conclusion that higher frequencies of home cooking, a lower risk of cardiovascular disease. Overall, the writing is clear. It is a good manuscript.

One concern arises from the writing of the last sentence of the Abstract. This study is relatively restricted to one municipal place in Japan and the sample size is not big enough to draw a conclusion applicable to the whole country. This point is oversold and needs to be toned down.

Line 220.  >home cooking was associated with…

Round 2

Reviewer 1 Report

The authors have responded appropriately. This is an interesting paper with increased scientific soundness after their corrections. There are limitations to the study but they are stated. Worth considering

Author Response

Thank you for your pleasant comment. We hope that the paper is now suitable for publication in Nutrients.